# Ciliate mitoribosome illuminates evolutionary steps of mitochondrial translation

Victor Tobiasson[1,2], Alexey Amunts[1,2]*

[1]Science for Life Laboratory, Department of Biochemistry and Biophysics, Stockholm University, Solna, Sweden; [2]Department of Medical Biochemistry and Biophysics, Karolinska Institutet, Solna, Sweden

**Abstract** To understand the steps involved in the evolution of translation, we used *Tetrahymena thermophila*, a ciliate with high coding capacity of the mitochondrial genome, as the model organism and characterized its mitochondrial ribosome (mitoribosome) using cryo-EM. The structure of the mitoribosome reveals an assembly of 94-ribosomal proteins and four-rRNAs with an additional protein mass of ~700 kDa on the small subunit, while the large subunit lacks 5S rRNA. The structure also shows that the small subunit head is constrained, tRNA binding sites are formed by mitochondria-specific protein elements, conserved protein bS1 is excluded, and bacterial RNA polymerase binding site is blocked. We provide evidence for anintrinsic protein targeting system through visualization of mitochondria-specific mL105 by the exit tunnel that would facilitate the recruitment of a nascent polypeptide. Functional protein uS3m is encoded by three complementary genes from the nucleus and mitochondrion, establishing a link between genetic drift and mitochondrial translation. Finally, we reannotated nine open reading frames in the mitochondrial genome that code for mitoribosomal proteins.

**\*For correspondence:**
amunts@scilifelab.se

**Competing interests:** The authors declare that no competing interests exist.

## Introduction

Mitoribosomes are composed of a catalytic rRNA core, encoded in the mitochondrial genome, and an outer shell of mitoribosomal proteins. During evolution, genetic information has been transferred from mitochondria to the nucleus independently in different species, the current mitochondrial genomes are highly diverse (*Janouškovec et al., 2017*). Previous structural studies have reported atomic models of mitoribosomes from eukaryotic supergroups such as Holozoa, Holomycota (*Brown et al., 2014*; *Amunts et al., 2014*; *Greber et al., 2014*; *Amunts et al., 2015*; *Greber et al., 2015*; *Desai et al., 2017*), and Discoba (*Ramrath et al., 2018*), previously Excavata (*Adl et al., 2019*). These mitoribosomes translate only a few mRNAs, of which all but one code for hydrophobic membrane subunits of the oxygenic phosphorylation complexes. A preliminary evolutionary analysis of some of those structures showed that mitoribosomes have evolved and gained new functions through a combination of destabilizing changes in mitochondrial DNA coding for rRNA (*Petrov et al., 2019*) and neutral evolution (*Gray et al., 2010*; *Lukeš et al., 2011*). However, the current insight is limited because the analyzed systems reflect a relatively narrow sampling of mitochondrial genome diversity, and it remains unclear whether other dispersal evolutionary strategies exist. Recently, single-cell RNA sequencing unraveled unexpected diversity in mitochondrial genomes, implying that mitochondria-encoded soluble proteins are widespread (*Keeling and McCutcheon, 2017*; *Wideman et al., 2020*). Therefore, to generate data to understand the evolution and function of mitochondrial translation, new evidence representing a larger variation of species is needed (*Lukeš et al., 2018*).

To this end, we searched for a model organism using two main criteria: 1) evolutionary distance from the previously characterized supergroups and 2) coding capacity of the mitochondrial genome beyond the conventional hydrophobic membrane proteins. The search singled out *Tetrahymena thermophila*, a ciliate protist from the phylum Alveolata. The *T. thermophila* mitochondrial genome differs substantially from those previously characterized. It encodes 43 proteins, half of which are soluble, varying in length from 59 to 1344 amino acids. In addition, 22 have unknown functions and are annotated as *Ymf* genes (*Brunk et al., 2003*). A further rationale for investigating *T. thermophila* is that it is an established and accessible laboratory instrument for genetics (*Nanney and Simon, 1999*), as well as a source of fundamental discoveries in biology. These include induction of cell-division synchrony (*Scherbaum and Zeuthen, 1954*), discoveries of lysosomes (*Elliott and Bak, 1964*), peroxisomes (*Baudhuin et al., 1965*), linear mitochondrial DNA (*Suyama and Miura, 1968*), dynein (*Gibbons and Rowe, 1965*), catalytic RNA (*Kruger et al., 1982*), and the molecular characterization of telomeres (*Blackburn and Gall, 1978*). In addition, alveolates are infectious agents commonly involved in meat or pet trade (*Chambouvet et al., 2020*).

## Results and discussion

### New proteins and conserved rRNA

To characterize the *T. thermophila* mitoribosome, we determined its structure using cryo-EM at 3.30–3.67 Å (*Figure 1*, *Figure 1—figure supplement 1*, *Supplementary file 1*). The resulting 4.0-MDa model consists of 92 different proteins, including two bL12m dimers. Of these, 46 proteins are mitochondria-specific, while 27 are newly identified including 9 *Ymf*-encoded with previously unassigned functions (*Figure 1A*, *Figure 1—figure supplements 2* and *3*, *Supplementary file 1*, *Video 1*). The protein nomenclature is consistent with the previous structures, whereas additional proteins are named to avoid overlap. Most of the newly identified proteins are associated with the small subunit and distributed across the head and the two regions that we named as the back

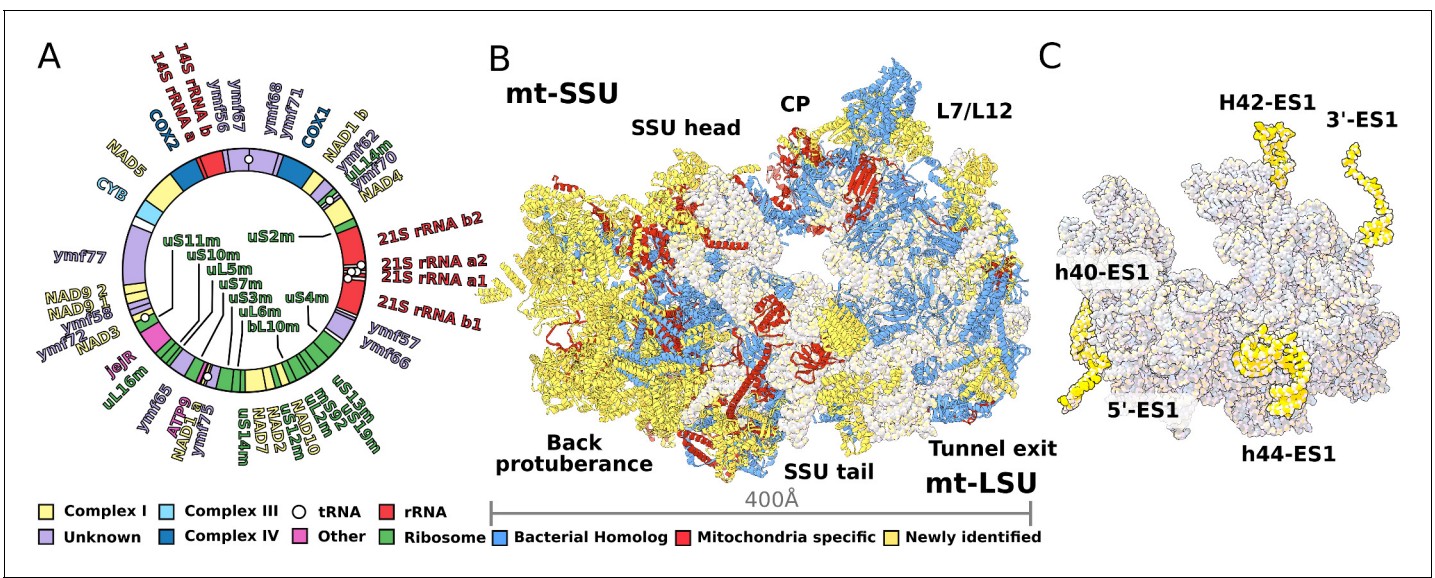

**Figure 1.** Structure of *T. thermophila* mitoribosome and newly identified proteins in the mitochondrial DNA. (**A**) Schematic representation of the mitochondrial genome of *T. thermophila* with newly identified proteins labeled inside the circle. (**B**) The overall structure of the mitoribosome showing mitochondria specific and newly identified proteins. (**C**) Mitoribosomal rRNA showing expansion segments (relative to *E. coli*) in yellow.

The online version of this article includes the following figure supplement(s) for figure 1:

**Figure supplement 1.** Electron microscopy data processing workflow.

**Figure supplement 2.** Positions of newly annotated proteins on the mitoribosome.

**Figure supplement 3.** Comparison of evolutionary conservation for mitoribosomal proteins.

**Figure supplement 4.** Secondary structure diagram of the *T. thermophila* LSU rRNA.

**Figure supplement 5.** Secondary structure diagram of the *T. thermophila* SSU rRNA.

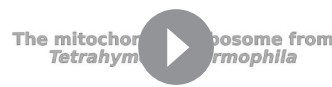

**Video 1.** Structure of the ciliate mitoribosome.
https://elifesciences.org/articles/59264#video1

protuberance and the body extension. These include SelR, a methionine-R-sulfoxide reductase (*Kryukov et al., 2002*) in the body extension, mS81 and an apo-frataxin protein (*Castro et al., 2019*) in the back protuberance, mS86. In the large mitoribosomal subunit (LSU), we identified a protein binding two iron-sulfur clusters via CDGSH motifs (*Sengupta et al., 2018*), mL107. Proteins in additional locations include mS93 bound to h44-ES1 in the small mitoribosomal subunit (SSU) tail, which we identified as the MTERF family of proteins. The MTERF family consists of four members featuring a 30-amino acid motif, containing leucine zipper-like heptads, reported to be involved in mitochondrial transcription as well as DNA replication (*Roberti et al., 2009*).

The rRNAs in both mitoribosomal subunits are split into two fragments (*Figure 1A*, *Figure 1—figure supplements 4* and *5*). However, the overall rRNA structure is conserved (*Figure 1C*), and expansion segments (ES) constitute only 3%, while 7% of the rRNA has been reduced. Some of those rRNA deletions are not structurally restabilized by proteins. For example, h56-59 of domain III is reduced to a single flexible h56 with no binding protein observed (*Figure 2*). The conservation of rRNA and the apparent absence of stabilizing protein elements imply that drivers other than rRNA reduction impact the mitoribosomal evolution.

## The mRNA channel is defined by split protein assembly uS3m encoded in different genomes

A fundamental stage in translation is the binding of mRNA to a dedicated channel, positioning the start codon of the open reading frame in the P-site to match the proper anticodon. In all the previously studied translation systems, including from organelles, the mRNA channel entrance is formed mainly by the conserved proteins uS3 and uS5.

In *T. thermophila*, we report that the assembly corresponding to uS3m is composed of three separate proteins encoded in the nucleic and mitochondrial genomes. The first protein, which was previously annotated as 'Ribosomal protein S3', partially corresponds to the N-terminal stabilizing domain (NTD) of the bacterial uS3 (*Figure 3*, *Figure 3—figure supplement 1*). We renamed it to mS92 as it constitutes the smallest fragment of uS3. The topology of the first ~50 amino acids is conserved; however, the similarity is broken by residues Tyr51-Tyr52-Tyr53 (*Alexandrov et al., 2012*) causing the mitochondrial structure to deviate from the original helical kink. The new conformation is then stabilized by the second protein mS31, with its short helix, as well as Lys57 via negatively charged rRNA phosphate. In the mitochondrial genome, we found the third protein encoded in the previously unknown *Ymf64*, which mainly corresponds to the functional C-terminal domain (CTD) of uS3 and forms a part of the NTD (*Figure 3*). We named it uS3m. The full NTD is then formed by two antiparallel β-strands from mS92 that are complemented by two β-strands from mS31 and an additional single β-strand from uS3m (*Figure 3*). The resulting β-sheet, therefore, consists of three different proteins assembled to make a bacterial-like uS3 NTD. In regard to the CTD, in Holozoa, it is completely deleted from the mitoribosome resulting in an expanded channel entrance (*Brown et al., 2014*), whereas in *T. thermophila* its genomic sequence is found to be split from the NTD. Assuming all ribosomes originally had intact uS3, the genomic splitting of uS3m reported here might represent a possible structural intermediate in the evolution of the mitoribosome toward the loss of the CTD observed in Holozoa. A similar splitting of the mitoribosomal protein uL2m has been recently reported in plants (*Waltz et al., 2020*).

The involvement of two genomes to produce a functional and stable protein further indicates an evolutionary link between genetic drift and mitochondrial translation. Our experimental finding therefore illustrates the requirement of tight mito-nuclear coevolution to maintain mitochondrial activities (*Shtolz and Mishmar, 2019*; *Mishmar, 2020*).

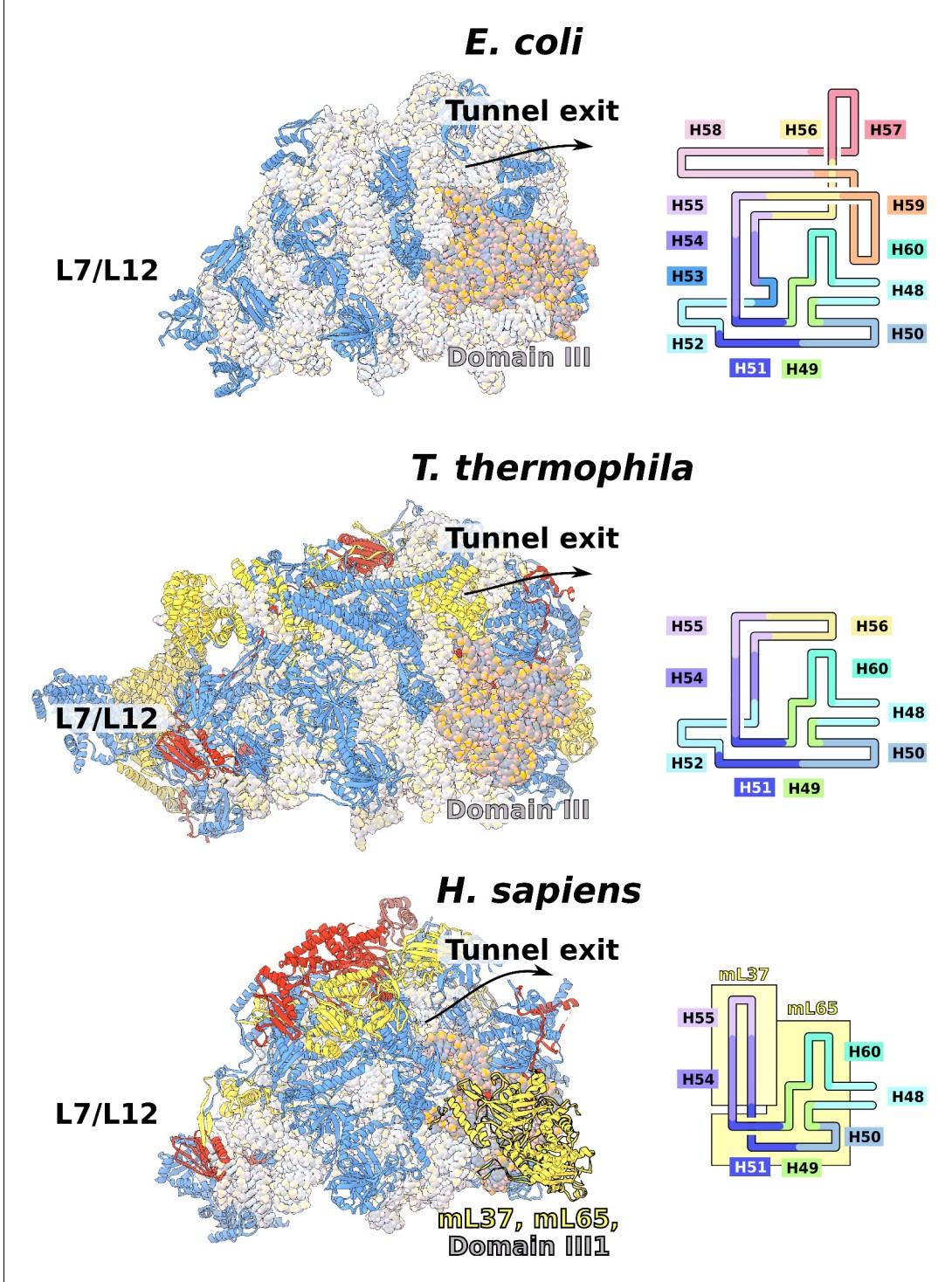

**Figure 2.** Evolution of rRNA domain III reduction. Comparison of H48-60 rRNA between *E. coli* and mitochondria of *T. thermophila* and *H. sapiens*. Left, LSU model with rRNA is shown in gray and domain III is highlighted, conserved proteins are in blue, shared mitoribosomal proteins are in red, and specific mitoribosomal proteins are in yellow. Right, schematic representation of the rRNA region subjected to reduction. In *H. sapiens,* the rRNA deletions are structurally restabilized by proteins mL37 and mL65. In *T. thermophila*, the rRNA reduction is less severe, and flexible elements such as H56 have no binding protein partners, suggesting an intermediate stage between *E. coli* and *H. sapiens*.

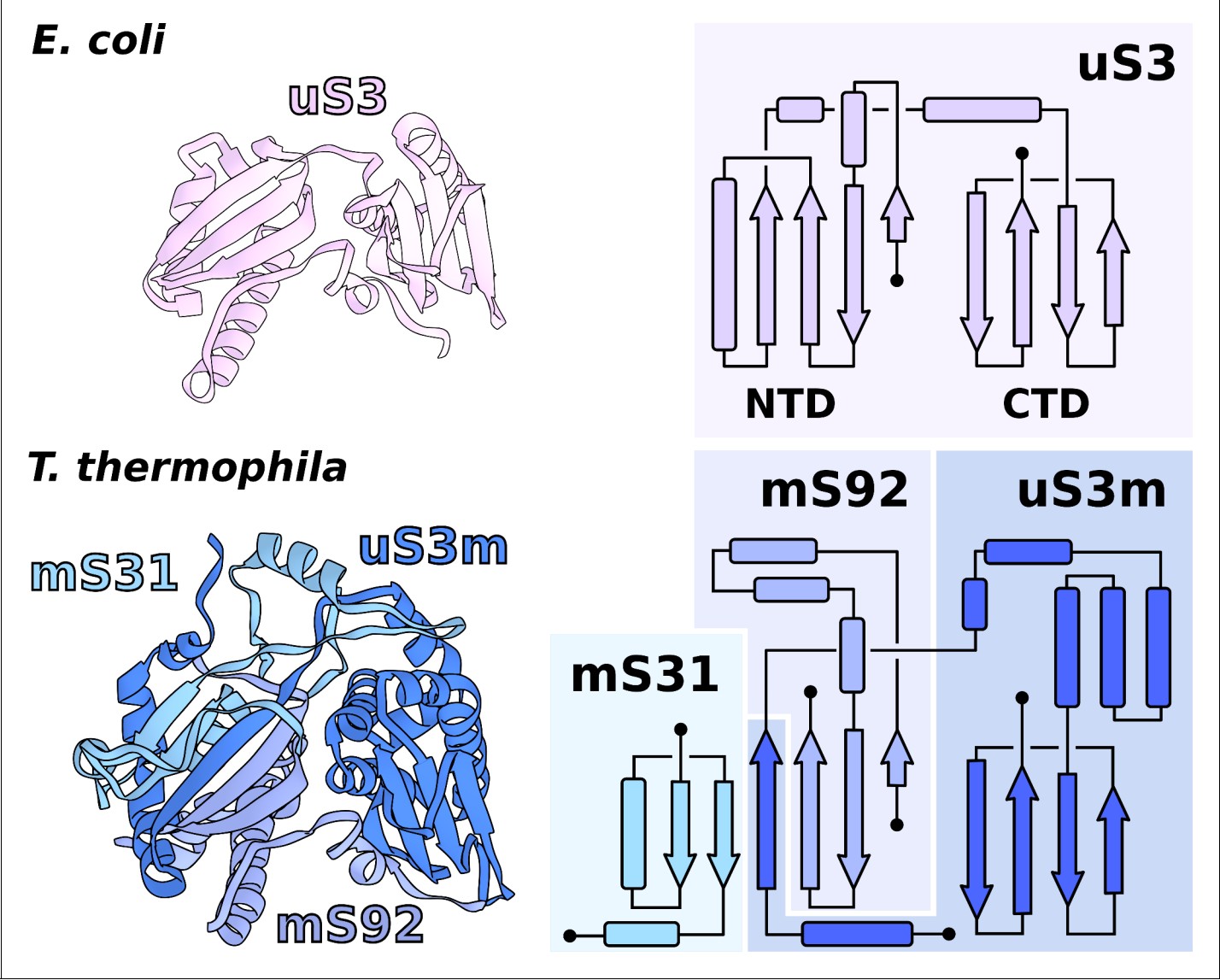

**Figure 3.** Functional uS3m consists of three separate proteins encoded in the nuclear and mitochondrial genomes. Comparison of uS3 between *E. coli* ribosome and *T. thermophila* mitoribosome. The topology diagram of uS3 shows the organization of its domains that are replaced by three different proteins in *T. thermophila*. Mitoribosomal mS31 (nuclear encoded), mS92 (nuclear encoded), and one strand from uS3m (mitochondria encoded) collectively correspond to uS3-NTD, whereas mitoribosomal uS3m corresponds to uS3-CTD.

The online version of this article includes the following figure supplement(s) for figure 3:

**Figure supplement 1.** Local resolution and density for the uS3m module.

## The expanded SSU head is connected to the body by an extensive protein network

The SSU head plays a key role during the process of translation elongation in promoting tRNA translocation. Flexible interactions with the body allow uncoupling of the head–body movement that is necessary for the sequential conformational changes during the translation cycle (*Ratje et al., 2010*).

The *T. thermophila* SSU head has seven extra proteins (mS29, 31, 33, 35, 75, 81, 89), while the bacterial homologs are extended by ~40%. In addition, two sites enriched with mitochondria specific proteins are identified on the SSU solvent-facing side. The first is the back protuberance that consists of eleven proteins (mS45, 47, 78-CTD, 83–88, 91, 92) facing away from the LSU binding interface; the second is the body extension that consists of eight proteins (mS23, 26, 37, 76, 77, 78-NTD, 79, uS11-NTD) protruding from the tRNA E-site (*Figure 1B*, *Figure 4A*, *Video 1*). The two moieties

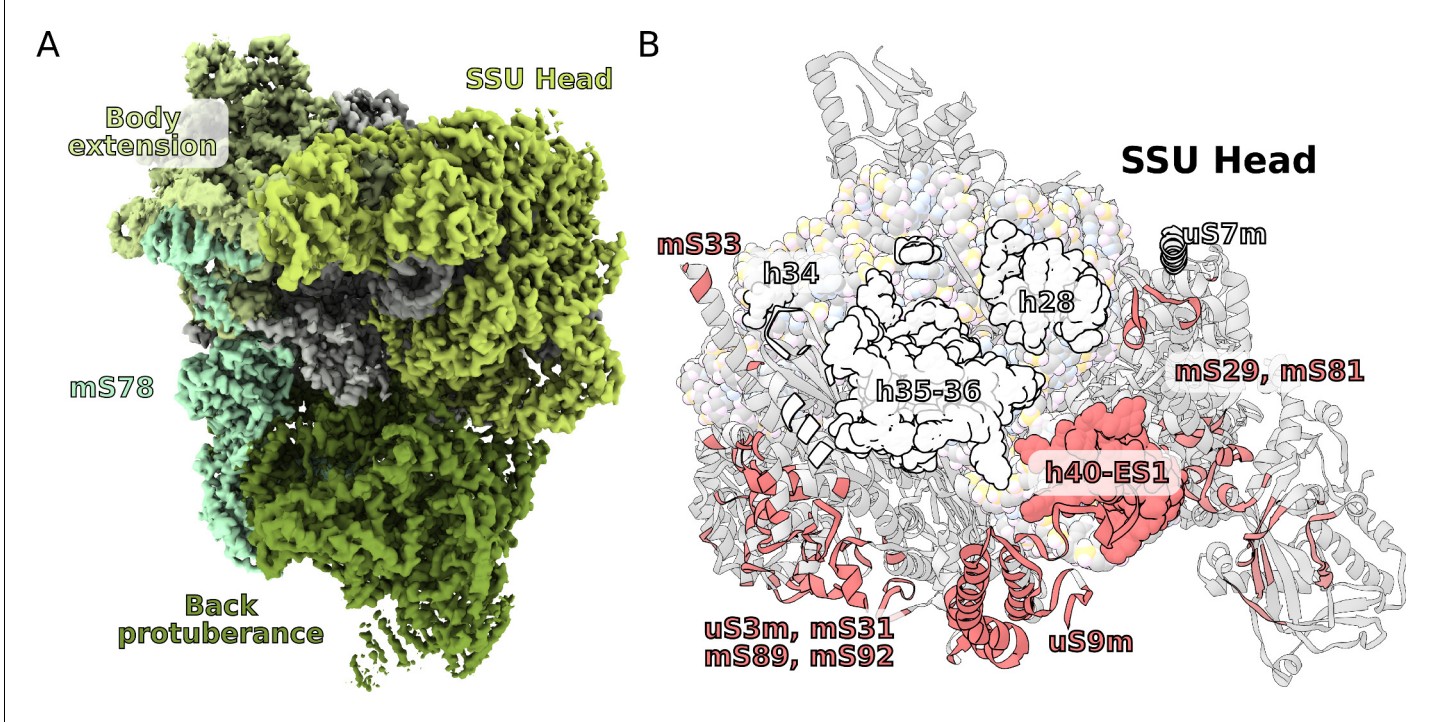

**Figure 4.** Unique functional features provide a distinct architecture of the mitoribosomal subunits. (A) The overall structure of the SSU is substantially affected by the back protuberance and body extension that are bound to the head and interconnected through mS78. (B) The contact area between the SSU head and body is illustrated on the head. Mitochondria specific contacts are shown in red, and conserved contacts are shown in white.

The online version of this article includes the following figure supplement(s) for figure 4:

**Figure supplement 1.** SSU head interactions with body extension.

appear to be interconnected with each other through mS78, which is the largest protein in our structure with 1509 modeled amino acids (*Figure 4A*, *Supplementary file 2*). The protein mS78 is divided into three domains: 1) NTD forming an extensive helical network with mS23, which builds the periphery of the body extension; 2) a central helical repeat linking the two new sites; and 3) CTD consisting of 34 helices capping the back protuberance.

The structures of the back protuberance and body extension indicate altered functions of the SSU. While the back protuberance coincides with putative RNA polymerase binding sites (*Demo et al., 2017*; *Kohler et al., 2017*), the body extension excludes bS1, which is considered to be one of the most ancient and conserved ribosomal proteins with functions involved in unfolding mRNAs for active translation (*Qu et al., 2012*). In addition, both protrusions interact with the head (*Figure 4A*). The back protuberance and the head are connected by nine proteins (mS31, 45, 47, 85, 87, 88, 89, 92, and uS9m) and h40-ES1. The body extension and the head are connected by four proteins (mS23, 26, 29, and 37). Additionally, mitochondria-specific extensions of uS9m, mS23, 29, and 31 form multiple interactions linking the head to the body (*Figure 4—figure supplement 1*). As a result, the buried surface area between the SSU head and body is almost doubled compared to that of bacteria: ~40,000 Å² vs ~21,000 Å² (*Figure 4B*).

Consistently with the altered structure of the head, the tRNA binding sites have also been remodeled (*Figure 5*). The C-terminal extension of uS13m presents a positively charged surface replacing the A-site finger. A newly identified protein mL102 makes further interactions with the A-site tRNA acceptor stem by extending its C-terminus downward from between the CP and 7/12 stalk. At the P-site, mL40 extends with its positively charged N-terminal helix toward the tRNA elbow. Taken together, the increased surface area between the head and body and the positioning of mitochondrial protein elements facing the A- and P-sites suggest altered interactions with the ligands.

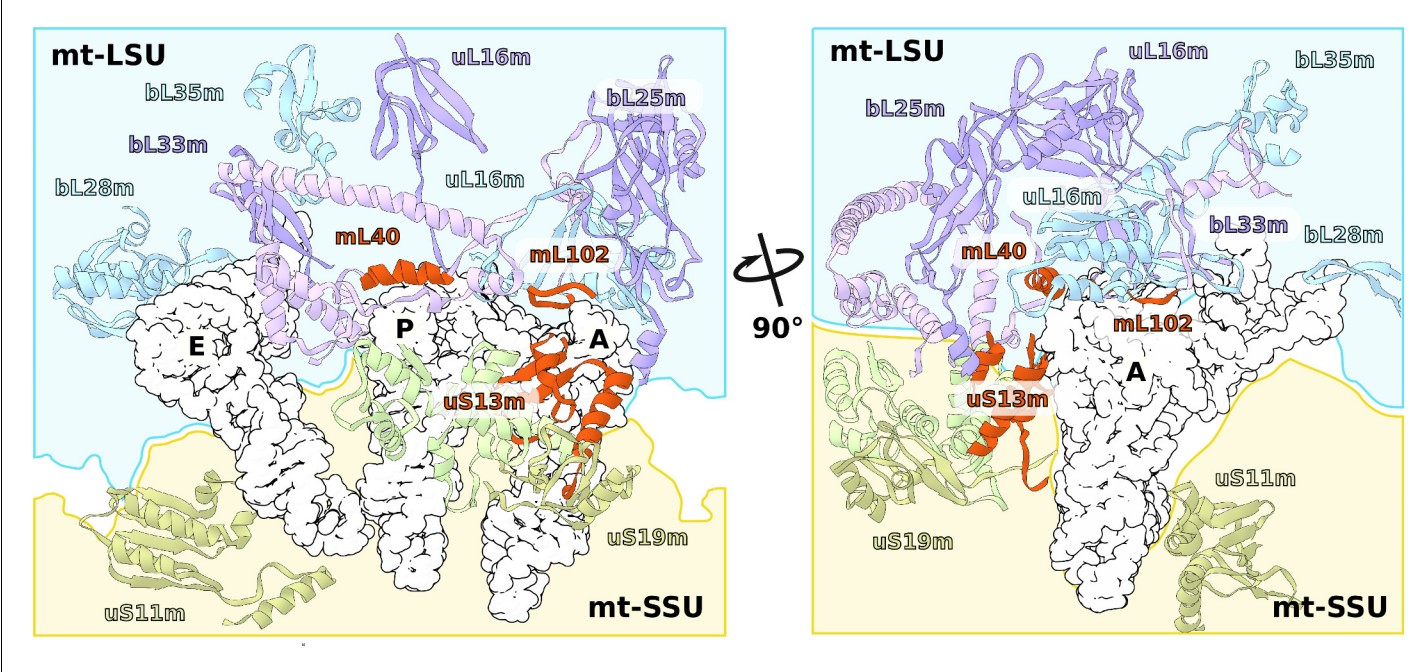

**Figure 5.** Specific protein elements interacting with tRNA binding sites. The conventional tRNA binding sites are indicated in white based on the canonical L-shape of tRNAs (PDBID: 5MDZ). Related proteins of the LSU and SSU are shown in blue/purple and white, respectively. Mitochondria specific elements encasing the tRNA binding sites are shown in red.

## Minimal central protuberance lacking 5S rRNA is a potential evolutionary intermediate

The central protuberance (CP) is a ubiquitous element of the LSU that forms bridges with the SSU head. Unlike cytosolic ribosomes that contain 5S rRNA as a core component of the CP, in mitochondria, the 5S rRNA has been replaced by other RNAs or protein. Analysis of the previously reported mitoribosomal structures concluded that the replacing elements, such as a Holozoan tRNA encoded between the two rRNA genes in the mitochondrial genome, could be incorporated into the CP to restabilize its core (*Petrov et al., 2019*).

In the *T. thermophila* mitoribosome, although the 5S is missing, no RNA or any substantial protein replacement is found (*Figure 6*, *Video 1*). This is despite a tRNA being encoded between rRNA genes, as in Holozoa (*Figure 1A*). The positions of the 5S C- and D-loops are partially occupied by a small protein loop of bL21m and an N-terminal extension of bL27m, respectively. Yet, the overall architecture of the CP remains intact, and the peripheral proteins are arranged similarly to those of other mitoribosomes. Also on the rRNA level, only a minor deviation from bacteria is found, which is in the 15-nucleotide long tip of H84 interacting with uL5m. The protein uL5m is traditionally associated with the 5S rRNA and is heavily reduced in our structure. Intriguingly, uL5m is the first reported mitoribosomal homolog that appears to be reduced. No other proteins are seen stabilizing the CP and the density suggests it is flexible.

Therefore, the *T. thermophila* mitoribosome has a minimal and the most flexible known CP, indicating a less significant role of LSU in the stabilization of the enlarged SSU head. These structural changes might reflect an intermediate evolutionary step after the loss of 5S rRNA and before the acquisition of a compensating structural replacement.

## Native L7/L12 stalk with four bL12m copies

The L7/L12 stalk mediates interactions with translational factors and is organized in three structural regions: the rRNA base (H42-44 rRNA), a protein linker (α8 of uL10, dimers of bL12 NTDs), and a mobile factor-recruiting domain (bL12 CTDs). Due to its high flexibility, the stalk has not been well

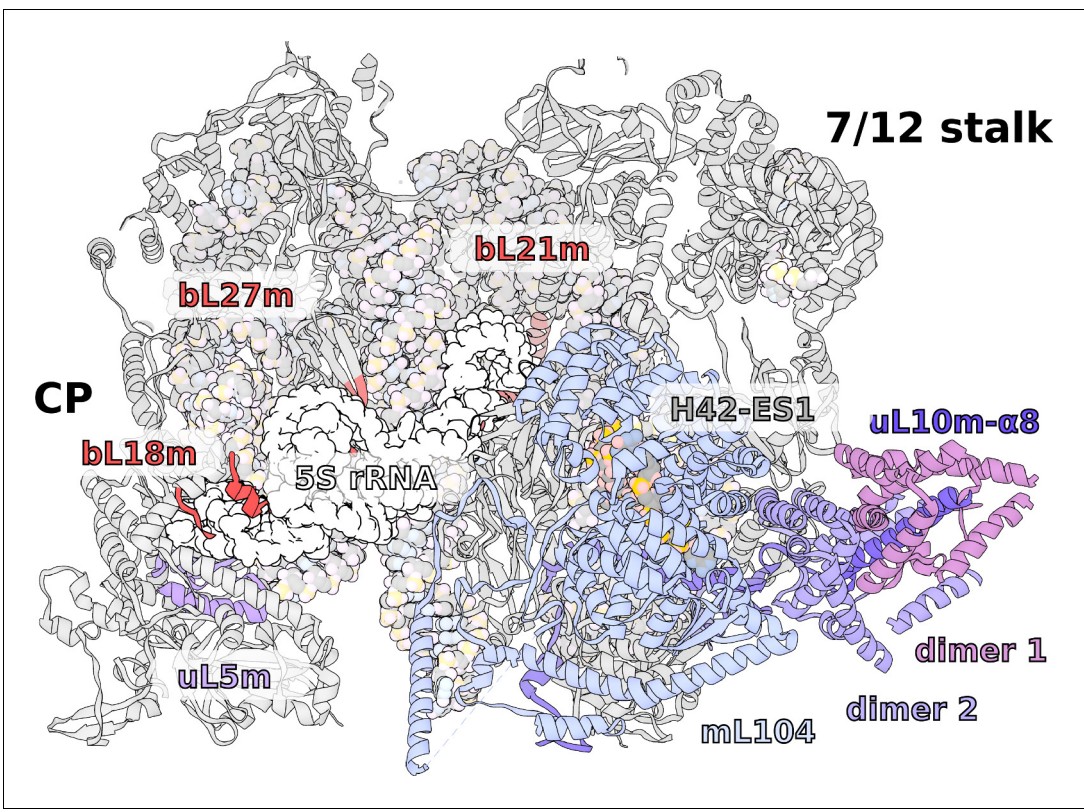

**Figure 6.** Specific features of the LSU. The LSU central protuberance lacks 5S rRNA, and the L7/L12 stalk consists of only two dimers. Superposition of *E. coli* 5S rRNA reveals no substantial protein replacement, apart from minor elements shown in red. The model of the native L7/L12 stalk reveals unusual conformation due to the presence of rRNA expansion H42-ES1 and mL104.

The online version of this article includes the following figure supplement(s) for figure 6:

**Figure supplement 1.** Structure of the L7/L12 stalk.

resolved in the previous cryo-EM reconstructions (*Brown et al., 2014*; *Amunts et al., 2014*; *Greber et al., 2014*; *Amunts et al., 2015*; *Greber et al., 2015*; *Desai et al., 2017*). However, a computational analysis of the mitoribosomal stalk predicted six bL12m copies arranged in three dimers (*Davydov et al., 2013*).

Our data shows a well-resolved linker domain allowing for modeling of the native uL10m and bL12m dimers (*Figure 6*, *Video 1*). In contrast to the prediction, only four bL12m copies arranged in two dimers fit the density. For uL10m we found a matching sequence encoded in the mitochondrial *Ymf74* sequence. Superimposition with the bacterial stalk revealed that the linker domain α8 of uL10m is straight and rigid, lacking the representative kinks that define its bL12-binding capacity in other ribosomes (*Liljas and Sanyal, 2018*; *Figure 6—figure supplement 1*). The structural basis for the stalk rigidity originates in a 25-nucleotide expansion H42-ES1 in the rRNA base. It is bound by the 90 kDa helical repeat protein mL104. This forms a stabilizing interface for the proximal bL12m dimer (*Figure 4—figure supplement 1*). As a result, the protrusion of the stalk is more distinct, placing the mobile factor-recruiting domains of bL12 CTDs further away from the A-site (*Figure 1B*, *Figure 6*). Such an arrangement provides a mechanical constraint on the number of bL12m copies within functional distance to test aa-tRNAs to a codon in the A-site and rationalizing the presence of only two bL12m dimers.

## Intrinsic protein visualized in the tunnel suggests putative targeting

Previous structural studies have focused on mitoribosomes that almost exclusively synthesized transmembrane proteins of the oxidative phosphorylation chain. Therefore, a translation-independent

membrane targeting was indicated (*Pfeffer et al., 2015*; *Englmeier et al., 2017*). In many other lineages of eukaryotes, multiple soluble proteins are encoded in mitochondria, however, whether a separate targeting system similar to the cytosolic translation apparatus exists is not known.

Since in the cytosol, ribosomes engage a universally conserved targeting system called the signal recognition particle (SRP), we carefully inspected the region on the mitoribosome that is equivalent to the SRP accessory proteins' binding sites. We observed a globular density near the tunnel exit, in the vicinity of H7, extending ~40 Å from the mitoribosome surface (*Figure 7A*). The local resolution of 3.2–5.0 Å in this region allowed modeling 150 residues arranged in eight helices, which belong to a nuclear-encoded protein that we named mL105 (*Figure 7B*, *Figure 7—figure supplement 1*). We found that this protein is annotated as 'signal peptide-binding domain protein fragment'. The topology and sequence of mL105 suggest homology to the M-domain of the bacterial SRP binding protein Ffh in its ribosome-bound form, while it lacks the GTP binding NG-domain (*Figure 7C*). The C-terminal helix of mL105 inserts into the tunnel and forms interactions with the hairpin loop of bL23m (*Figure 7A*). Moreover, it would inevitably contact the nascent chain. A similar functional insertion of Ffh was suggested based on the low resolution reconstruction of the reconstituted cytosolic system (*Jomaa et al., 2016*) and confirmed by cross-linking data (*Denks et al., 2017*). We therefore report an intrinsic feature of the ciliate mitoribosome that provides the first evidence for a putative protein targeting in mitochondria.

## Conclusions

Although the requirement for proteins synthesized in mitochondria is vital, the underlying mechanisms were previously only partially explored in the isolated species. The current structural analysis

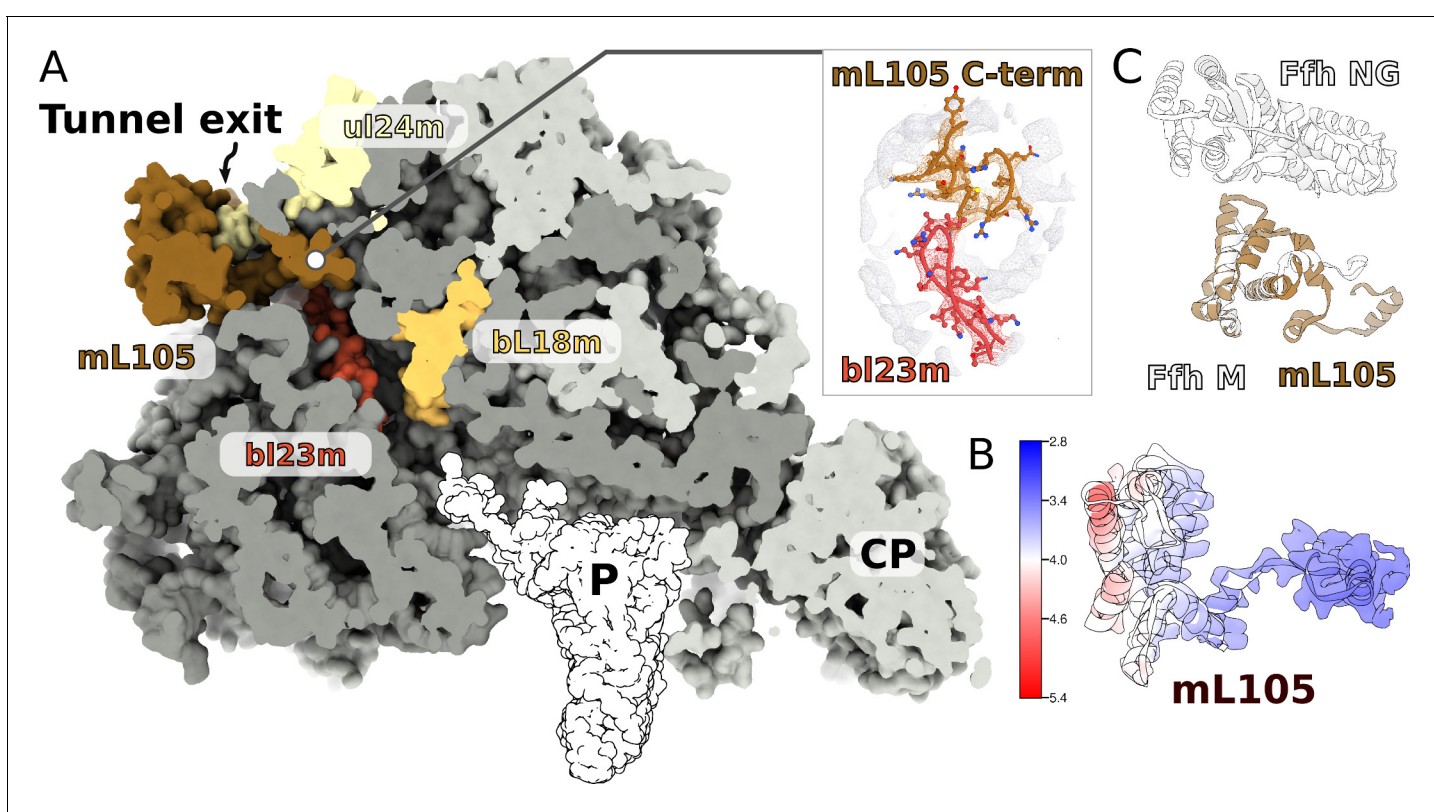

**Figure 7.** A signal peptide-binding domain protein is bound to the LSU. (**A**) Slicing through the LSU shows the tunnel path and the targeting protein mL105 bound at the tunnel exit. The protein mL105 is positioned in a way that would affect the nascent polypeptide path. Inset displays interactions between mL105 CTD and bL23m close to the tunnel exit. (**B**) Model and density for mL105 colored by local resolution. (**C**) Superposition of the ribosome-bound Ffh M-domain (PDBID: 5GAF) with mL105 shows structural similarity.

The online version of this article includes the following figure supplement(s) for figure 7:

**Figure supplement 1.** Local resolution density for mL105.

of the ciliate mitoribosome, which is evolutionarily distant from the previously characterized, shows that mitochondria have evolved independent features related to all functional aspects of translation. The data revealed extra proteins on the SSU that might affect the conformational landscape of the head movement, identified protein-rich tRNA binding sites, and reported prime evidence for a protein targeting system through a signal peptide-binding domain protein found at the LSU tunnel exit.

From the evolutionary perspective, a surprising feature is that despite the multiple additional proteins that form a distinct overall structure, the rRNA is generally conserved. This suggest evolutionary drivers other than rRNA for mitoribosomal diversity. In addition, our finding that rRNA deletions are not structurally restabilized by proteins imply that the ciliate mitoribosome represents an evolutionary intermediate between other eukaryotic supergroups, providing a reference point for further investigation of the development of translation. Finally, we report a functional protein composed of three separate proteins encoded in the nuclear and mitochondrial genomes, providing experimental link between genetic drift and mitochondrial translation.

Together, the distinct *T. thermophila* mitoribosome structural model illustrates the functional diversity of mitochondrial translation and provides a framework for examination of its evolution.

## Materials and methods

### Strains and growth conditions

*T. thermophila* SB210 cells (from ATCC) were cultured axenically in 1% (w/v) yeast extract, 2% proteose peptone supplemented with 3% (v/v) glycerol at 36°C with 80 rpm shaking. Upon reaching mid log phase (~$8\times10^5$ cells/ml), the cells were harvested by centrifugation for 10 min at 1300 g. All the subsequent steps were performed at 4°C. The pellets were resuspended in a minimal volume of homogenization buffer (20 mM HEPES KOH pH 7.5, 350 mM mannitol, 5 mM EDTA). Lysis was achieved by homogenization in a Dounce homogenizer with 30 strokes of a tightly fitting plunger. Debris were removed by two centrifugations at 800 g for 10 min. Mitochondria were then pelleted by centrifugation at 7000 g for 20 min. The pellets were resuspended in minimal volume and loaded onto a discontinuous sucrose gradient (20 mM HEPES KOH pH 7.5, 1 mM EDTA with 15, 23, 32%, and 60% sucrose) and centrifuged at 90,000 g for 1 hr. Mitochondria were collected between the 60% and 32% sucrose layers and flash frozen in aliquots of 2 mL in liquid nitrogen.

### Purification of mitoribosomes

All the subsequent steps were carried out on ice. About 100 mg of mitochondrial protein was thawed and solubilized by adding 5 volumes of Triton buffer (25 mM HEPES KOH pH 7.5, 20 mM KCl, 25 mM MgOAc, 1.7% Triton X-100 and ROCHE complete EDTA-free protease inhibitor tablets), followed by a 15 min incubation on a rolling mill for 15 min. Mitochondrial membranes were removed by centrifugation at 30,000 g for 20 min. The released mitochondrial proteins were transferred onto 1 M sucrose cushion (25 mM Hepes KOH pH 7.5, 20 mM KCl, 15 mM MgOAc, 1% Triton X-100, and 1 M sucrose), and mitochondrial ribosomes pelleted at 235,000 g for 4 hr. The pellet was resuspended in a minimal amount of buffer (25 mM HEPES KOH pH 7.5, 20 mM KCl, 15 mM MgOAc). Aggregates were removed by centrifugation for 5 min at 14,000 g. The clarified supernatant was transferred to linear 15–30% (w/v) sucrose gradients. Gradients were centrifuged at 90,000 g for 16 hr and fractionated into 400 µL fractions. Fractions with an enriched absorbance at 260 nm were pooled and centrifuged at 235,000 g for 60 min to pellet mitoribosomes. Final pellets were resuspended in the same buffer without sucrose.

### Cryo-EM and model building

Quantifoil R2/2 300 mesh copper grids were coated with a ~3 nm continuous carbon support produced in house and glow discharged for 30 s at 25 mA. About 3 µL of the purified ribosomal sample at $OD_{260}$ 3.0 was applied and incubated for 30 s at 100% humidity at 4°C before blotting for 3 s and plunge frozen into liquid ethane using an FEI Vitrobot MkIV. The cryo-EM data were collected using a Titan Krios microscope operated at a voltage of 300 kV and equipped with a K2 Summit detector (Gatan). Data were collected at a nominal magnification of 130,000x corresponding to a pixel size of 1.07 Å$^2$/px, and a fluence of 30 electrons divided into 20 dose fractions at a flux of 5 e$^-$/Å$^2$s and a

final defocus range from −0.5 to −3.0 μm. A total of 5511 micrographs were selected after manual screening based on CTF fitting and overall qualitative appearance.

Initial motion correction was carried out using motioncor2 version 2.1 (*Zheng et al., 2017*), followed by CTF estimation by Gctf version 1.06 (*Zhang, 2016*). All further processing was done with RELION 2.7 and 3.0 (*Zivanov et al., 2018*). The workflow of the data processing is shown in *Figure 1—figure supplement 1*. A total of 450,250 particles were picked using Gaussian blobs and extracted with a box of 550 pixels. The box was binned to 128 pixels for the following classification steps. After several rounds of 2D classification, 133,650 particles were left for the 3D classification. After 3D classification, three classes with good quality were selected for the final reconstruction. In total, 99,380 particles were merged for 3D refinement. The final resolution of the 3D auto-refinement after post-processing was 3.67 Å. Application of masks for the LSU and SSU during refinement further improved the resolution of these regions to 3.38 Å and 3.61 Å, respectively. We also applied local masks to the L7/L12 stalk, CP, SSU head, back protuberance, which resulted in improved quality of local maps with resolutions ranging between 3.30 Å and 3.61 Å. All the resolutions were estimated with the gold-standard Fourier shell correlation 0.143 criterion with high-resolution noise substitution. All the local resolution maps were calculated using RELION 3.0 (*Zivanov et al., 2018*).

Model building was done in *Coot* 0.8.9.2 (*Emsley et al., 2010*). Initially, models of the mitoribosome from *S. cerevisiae* (PDB ID: 5MRC) and *H. sapiens* (PDB ID: 3J9M) were fitted to the map and served as protein backbone and rRNA references. Most of the proteins were built de novo using a combination of bulky side chain patterns, fold identification by PDBeFold (*Krissinel and Henrick, 2004*), mass-spec data, and assigning putative primary sequence followed by BLAST (*Altschul et al., 1990*) searches. The model was initially built and refined against a composite map consisting of all six masked regions and finally refined and validated using the consensus map. All models were refined iteratively using PHENIX (*Liebschner et al., 2019*) realspace refinement and validated using MolProbity (*Williams et al., 2018*). The data collection, model refinement and validation statistics are presented in *Supplementary file 1*. All figures were generated using either Chimera (*Pettersen et al., 2004*) or ChimeraX (*Goddard et al., 2018*) with annotations and vector editing was done using Inkscape.

## Acknowledgements

The authors thank the SciLifeLab cryo-EM and mass spectrometry facilities, and G von Heijne for his comments on the manuscript. This work was supported by the Swedish Foundation for Strategic Research (FFL15:0325), Ragnar Söderberg Foundation (M44/16), Swedish Research Council (NT_2015–04107), Cancerfonden (2017/1041), European Research Council (ERC-2018-StG-805230), Knut and Alice Wallenberg Foundation (2018.0080), EMBO Young Investigator Program. The cryo-EM facility is funded by the Knut and Alice Wallenberg, Family Erling Persson, and Kempe foundations.

## Additional information

### Funding

| Funder | Grant reference number | Author |
|---|---|---|
| Ragnar Söderbergs stiftelse | M44/16 | Alexey Amunts |
| Cancerfonden | 2017/1041 | Alexey Amunts |
| H2020 European Research Council | ERC-2018-StG- 805230 | Alexey Amunts |
| Knut och Alice Wallenbergs Stiftelse | 2018.0080 | Alexey Amunts |
| European Molecular Biology Organization | EMBO Young Investigator Program | Alexey Amunts |
| Swedish Foundation for Strategic Research | FFL15:0325 | Alexey Amunts |
| Swedish Research Council | NT_2015–04107 | Alexey Amunts |

The funders had no role in study design, data collection and interpretation, or the decision to submit the work for publication.

## Author contributions
Victor Tobiasson, Alexey Amunts, Conceptualization, Data curation, Formal analysis, Validation, Investigation, Visualization, Methodology, Writing - original draft, Writing - review and editing

## Author ORCIDs
Victor Tobiasson [ID] https://orcid.org/0000-0001-8920-017X
Alexey Amunts [ID] https://orcid.org/0000-0002-5302-1740

## Decision letter and Author response
Decision letter https://doi.org/10.7554/eLife.59264.sa1
Author response https://doi.org/10.7554/eLife.59264.sa2

# Additional files

## Supplementary files
- Supplementary file 1. Cryo-EM data collection, refinement and validation statistics.
- Supplementary file 2. Summary of the mitoribosomal proteins.
- Transparent reporting form

## Data availability
The electron density maps have been deposited into EMDB, with accession codes EMD-11032 (monosome), EMD-11033 (LSU), EMD-11034 (SSU), EMD-11035 (CP), EMD-11036 (L7/L12 stalk), EMD-11037 (head), EMD-11038 (back protuberance). The model has been deposited in the PDB, with accession code 6Z1P.

The following datasets were generated:

| Author(s) | Year | Dataset title | Dataset URL | Database and Identifier |
|---|---|---|---|---|
| Tobiasson V, Amunts A | 2020 | monosome | https://www.ebi.ac.uk/pdbe/entry/emdb/EMD-11032 | Electron Microscopy Data Bank, EMD-11032 |
| Tobiasson V, Amunts A | 2020 | LSU | https://www.ebi.ac.uk/pdbe/entry/emdb/EMD-11033 | Electron Microscopy Data Bank, EMD-11033 |
| Tobiasson V, Amunts A | 2020 | SSU | https://www.ebi.ac.uk/pdbe/entry/emdb/EMD-11034 | Electron Microscopy Data Bank, EMD-11034 |
| Tobiasson V, Amunts A | 2020 | CP | https://www.ebi.ac.uk/pdbe/entry/emdb/EMD-11035 | Electron Microscopy Data Bank, EMD-11035 |
| Tobiasson V, Amunts A | 2020 | L7/L12 stalk | https://www.ebi.ac.uk/pdbe/entry/emdb/EMD-11036 | Electron Microscopy Data Bank, EMD-11036 |
| Tobiasson V, Amunts A | 2020 | head | https://www.ebi.ac.uk/pdbe/entry/emdb/EMD-11037 | Electron Microscopy Data Bank, EMD-11037 |
| Tobiasson V, Amunts A | 2020 | back protuberance | https://www.ebi.ac.uk/pdbe/entry/emdb/EMD-11038 | Electron Microscopy Data Bank, EMD-11038 |
| Tobiasson V, Amunts A | 2020 | Model | https://www.rcsb.org/structure/6Z1P | RCSB Protein Data Bank, 6Z1P |

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
