## [Decision Letter]

[Editors' note: this paper was reviewed by Review Commons.]

**Acceptance summary:**

The cryo EM structure of the *Tetrahymena thermophila* mitocondrial ribosome reveals several surprising differences from the human and yeast, thus shedding light on the evolution of mitoribosomes. The splitting of conserved ribosomal protein, S3, into three separate proteins encoded in the nucleus highlights the remarkable co-evolution of mitochondrial and nuclear genomes. The authors were also able to annotate nine mitochondrial genes of previously unknown function, which they found encoded mitoribosomal proteins.

---

## [Author Response]

We wish to thank all the reviewers and the editor for dedicating their time to assess our manuscript and bringing up a number of minor points regarding data presentation and specific formulations in the text, each one of which has been implemented as described below. Three supplementary figures and a video have been added to the manuscript in order to better illustrate the additional points.

All the reviewers agreed that the work is an important contribution to the field of protein synthesis, the manuscript is well written, and the quality of the figures is high. Therefore, we hope the editor will find that the comments have been adequately addressed and consider the revised version to be acceptable for publication in *eLife*.

Reviewer #1 (Evidence, reproducibility and clarity):This manuscript described the structure its authors have obtained of the mitochondrial ribosomes from Tetrahymena by cryo-EM. The procedures the authors used appear to be state of the art, and the resolution of the EM maps they produced is sufficient to support the structure the authors have derived from them. The manuscript is well-written, and illustrations set is unusually good.

Thank you.

The only part of the manuscript that would benefit from further thought is the first sentence of the Abstract, which this reviewer did not understand. If the authors cannot think of a way to fix it, they might consider deleting it altogether. Otherwise, this manuscript is ready for submission.

Changed to: “Evolution of mitochondrial translation reflects transition steps in cellular development of eukaryotic species.”

Reviewer #1 (Significance):Compared to bacterial ribosomes or to those found in the cytoplasms of eukaryotes, little is known about the structure of mitochondrial ribosomes and about the details of how they function. A handful of structures have already been obtained for mitochondrial ribosomes by cryo-EM, and what they have taught us is how different they are from cytoplasmic ribosomes, and much they vary from one species to the next. The authors of this manuscript elected to determine the structure of mitochondrial ribosomes obtained from a species that is unrelated to any used for similar studies in the past, and they have been rewarded for doing so. The mitochondrial ribosomes in Tetrahymena differ a lot from those found in yeast and mammals. This manuscript will appeal to everyone interested in protein synthesis, and ribosome structure. Evolutionary biologist and cell biologists will also take note.(This reviewer is a structural biologist who has worked on ribosome structure and function for most of his career.)Reviewer #2 (Evidence, reproducibility and clarity):Tobiasson and Amunts describe a ~3.3 A cryo-EM structure of the mitochondrial ribosome from *T. thermophila*. The study reports several interesting findings (including plenty of previously unreported mito-proteins) and brings new insights into the evolution of mitoribosomes. For example, essential ribosomal protein S3 (conserved from bacteria to metazoans) consists of three separate proteins in T. t. mitoribosomes, two of which are encoded by nuclear genome, representing a spectacular example of co-evolution of mitochondrial and nuclear genomes. Furthermore, the structure reports an "expanded" head of the small subunit, unique features of the large subunit (e.g. a minimalistic central protuberance), and a putative protein-targeting system that is homologous to the bacterial SRP-binding protein Ffh. The quality of the structure is high, and the manuscript is clearly written.

Thank you.

I suggest a few minor comments and edits:Figure 1C – RNA expansions are shown relative to what organism?

Added “relative to *E. coli*”

"by a Tyr51-Tyr52-Tyr53 residues" – remove "a"

Removed.

In Figure 3 – label "NTD" and "CTD" and/or helices/sheets in bacterial uS3 structure, which are discussed in the main text.

Labeled.

Could the authors comment on whether there are other examples of heterooligomerization in mitoribosomes or protein splitting, whose complexity (intertwining) is similar to that of split S3 (including previously published mitoribosomes)?

Added with the corresponding reference: “A similar splitting of the mitoribosomal protein uL2m has been reported recently in plants (Waltz et al., 2020).”

"136 Assuming all ribosomes originally had intact uS3, the here reported genomic splitting of uS3m represents a structural intermediate in the evolution of the mitoribosome."It is not clear what this sentence suggests. Is evolution of mitoribosomes in *T. thermophila* related to evolution of mitoribosomes in other organisms? Do the authors speculate that mitoribosomes in other organisms that do not have uS3m split, would necessarily evolve through the splitting stage in the future or have already passed this stage?

As suggested by the reviewer, we added additional information to clarify, and the sentence now reads “splitting of uS3m might represent a possible structural intermediate in the evolution of the mitoribosome towards the loss of the CTD observed in Holozoa.”

The authors provide a nice comparison of part of LSU of the mitoribosome with that of *E. coli* ribosome (Figure 2). It might help the readers to also appreciate the difference between SSUs of the mito-ribosome and *E. coli*. Could a comparison panel be added to Figure 4?

As suggested by the reviewer, a comparison figure has been added as Figure 1—figure supplement 3.

The title and Discussion in section "The expanded SSU head is constrained by an extensive protein network" may confuse a reader by implying that the head dynamics is constrained. I suspect this is not what the authors imply, as head movement is conserved and essential for tRNA translocation. Given the essential role of head movement, it is unlikely that the additional proteins and RNA extensions inhibit or constrain head dynamics (e.g. by limiting the extent of head swivel). I suggest that the authors edit this title and section to better reflect their interpretation of the "expanded SSU head" and avoid the word "constrained", unless they provide experimental data showing that the head dynamics are indeed constrained.

As suggested by the reviewer, we deleted “constrained” from the title and the text body. This section has been also edited, where “function” is replaced with “structure”. In the Abstract, we deleted “movement”, and in conclusions “change”.

Several key findings of new proteins/interactions are not sufficiently illustrated by local density to allow readers appreciate whether the quality of experimental data supports structural interpretations. I suggest showing a region of local density (with side chains modeled) for:

– Split uS3

– mL105 (part of the view in Figure 7B, with side chains shown, would suffice)

As suggested by the reviewer, this information has been added as Figure 3—figure supplement 1 and as Figure 7—figure supplement 1, respectively.

Reviewer #2 (Significance):This is a significant study reporting the first cryo-EM structure of mitoribosomes from *Tetrahymena thermophila*. Several findings in this work demonstrate the unique features of cilate mitoribosomes and provide insights into co-evolution of the mitochondrial and nuclear genomes.Review signed by: Andrei A KorostelevReviewer #3 (Evidence, reproducibility and clarity (Required):In this manuscript, Tobiasson & Amunts describe the detailed molecular structure of the mitochondrial ribosome from *Tetrahymena thermophila*. Addressing the need to elucidate the great compositional, structural, and functional diversity of mitochondrial ribosomes and their differing adaptations to organellar translation, this is an important study. The structure illustrates new variants of how even functionally important sites, such as the mRNA binding channel, the tRNA binding sites, the interface between the small subunit head and body, and the exit tunnel, are modified and remodelled in mitochondrial ribosomes. The finding that one ribosomal protein is split into three genes, only for the three parts to then come together to form a fold that strongly resembles the original protein S3 is remarkable and underscores the complexity of mitoribosomal – or indeed mitochondrial – evolution.The paper is concise and well written, with each section discussing a functionally important and structurally unique element of the Tetrahymena mitoribosome. The paper extends our knowledge of mitochondrial translation and mitoribosome evolution and will be an important contribution to the field. I am providing some comments below that I believe should be addressed for completeness or context, but I have no major concerns that would adversely impact the suitability of the manuscript for publication.Major comments:None. The data support the conclusions and no further experimental work is necessary.Minor comments:Introduction – probably "neutral evolution" is better than "neutral selection".

Changed.

Subsection “The expanded SSU head is connected to the body by an extensive protein network” – "largest identified ribosomal protein": in Tetrahymena?

Added for clarification: “in our structure”

"Taken together… suggest altered dynamics during translation elongation." Do the authors think that the relative motions of head and body are different? Could the authors possibly find any information on this in their cryo-EM data, e.g. in their existing 3D classification or by multi-body refinement and subsequent variance analysis? Or do they think that the structural changes observed mostly rigidify the architecture instead of inducing new states?

We performed the analysis as suggested by the reviewer, and found no meaningful information, probably due to lack of ligands in this data set. Therefore, for clarification, the statements related to function have been toned down.

"Whether a separate targeting system similar to the cytosolic translation apparatus exists is not known." – does this refer to the existence of the SRP system in mitochondria in general, or just the case of Tetrahymena? In the latter case, this could be made more explicit (possibly because this is the structure of a mitoribosome that synthesizes large numbers of soluble proteins?).

We added the requested information: “In many other lineages of eukaryotes multiple soluble proteins are encoded in mitochondria, however whether a separate targeting system similar to the cytosolic translation apparatus exists is not known.”

Figure 7: The figure (panel C) shows that mL105 and the Ffh M-domain share a similar fold. Could the authors provide a structural comparison between mL105 and the Ffh M-domain when bound to the ribosome? Is the architecture of the complex conserved or is it different?

The structural comparison has been provided, and the conserved architecture is illustrated in Figure 7C. We added “ribosome-bound” in the text and in the Figure 7C legend.

"This suggest evolutionary drivers other than rRNA for mitoribosomal diversity." The evolutionary drivers are likely variable between lineages, with e.g. mammals exhibiting very different evolutionary trajectories compared to other organisms. Slightly extending the Discussion to incorporate these different pathways may result in a more inclusive view of the structural evolution of mitoribosomes across lineages and may highlight the novel contributions of this new structure on this topic.

This comment is related to one of the last sentences of the “Conclusions” section, which describes the implications of the reported structural data. To provide an overview of the structural data, we now added a supplementary video. We agree that studying evolution across different lineages would be very informative in the future, and hope that the structural data reported in this research article will be also useful in this regard.

Protein nomenclature: It is my understanding that the nomenclature was chosen such as to avoid overlap with the mitoribosomal proteins in all other existing structures (except for proteins shared between different mitoribosomes). Is this correct?

Thank you, we added a clarification: “The protein nomenclature is consistent with the previous structures, whereas additional proteins are named such as to avoid overlap.”

Figures: The figures are generally well made and the schematic depictions greatly help conceptualise the discoveries. One aspect for the authors to consider is their choice of a rendering style that is essentially devoid of depth cues (such as shadows) for many panels and can make it difficult to fully appreciate what is depicted in three dimensions, particularly in complex structures like Figures 1, 2, 4B. Introducing subtle changes may be worth considering.

Thank you for this valuable suggestion how to possibly improve the representation. To improve the illustration of the details in the context of the whole complex, we prepared a supplementary video overviewing each one of the discussed aspects. We also tried to follow the reviewer’s suggestion regarding the figures, where possible.

Validation: Model vs. map FSC curve should be added and resolution where the curve drops below 0.5 should be added to Supplementary file 1 (refinement table).

Added.

Reviewer #3 (Significance (Required)):Mitoribosomes exhibit extreme compositional and structural diversity, indicative of great evolutionary plasticity, which contrasts sharply with the otherwise relatively well conserved architecture of ribosomes within domains of life. Therefore, substantial efforts in the field are being devoted to understanding the structural, functional, and mechanistic ramifications of this diversity. These efforts provide insight into the functional specialisation of the mitoribosome, its unique structural features, and the mechanisms of molecular evolution that generated this diversity.The data presented here contribute new insight into all of these questions and represent a significant advance. As stated by the authors in the Introduction, this work in Tetrahymena provides a structural view of a mitoribosome synthesising a functionally diverse range of protein products, and the discovery of a built-in protein targeting machinery is a striking result. The findings reported in the paper will therefore be of great interest to the research communities studying ribosomes, molecular evolution, or mitochondrial biology, as well as to structural biologists in general.Own expertise: Electron microscopy, including structural biology of ribosomes.